# Role of Adjuvant Radiotherapy in Patients with Cervical Cancer Undergoing Radical Hysterectomy

**DOI:** 10.3390/jpm13101486

**Published:** 2023-10-12

**Authors:** María Alonso-Espías, Mikel Gorostidi, Myriam Gracia, Virginia García-Pineda, María Dolores Diestro, Jaime Siegrist, Alicia Hernández, Ignacio Zapardiel

**Affiliations:** 1Gynecologic Oncology Unit, La Paz University Hospital, 28046 Madrid, Spain; mariaalonsoespias@gmail.com (M.A.-E.); dra_gracia@hotmail.com (M.G.); virginia.garciapineda@gmail.com (V.G.-P.); mdtejeda@gmail.com (M.D.D.); jaimesiegrist@hotmail.com (J.S.); aliciahernandezg@gmail.com (A.H.); ignaciozapardiel@hotmail.com (I.Z.); 2Gynecologic Oncology Unit, Donostia University Hospital, 20014 San Sebastian, Spain

**Keywords:** cervical cancer, intermediate-risk factors, adjuvant radiotherapy

## Abstract

The benefit of adjuvant radiotherapy (RT) after radical hysterectomy in patients with cervical cancer remains controversial. The aim of this study was to determine adjuvant RT’s impact on survival in accordance with Sedlis criteria. Patients with early-stage cervical cancer undergoing radical hysterectomy between 2005 and 2022 at a single tertiary care institution were included. A multivariate analysis was performed to determinate if RT was an independent prognostic factor for recurrence or death. We also analysed whether there was a statistically significant difference in overall survival (OS) between patients who met only one or two Sedlis criteria, depending on whether they received adjuvant RT or not. 121 patients were included in this retrospective study, of whom 48 (39.7%) received adjuvant RT due to the presence of unfavourable pathological findings. In multivariate analysis, RT was not found to be a statistically significant prognostic factor for OS (*p* = 0.584) or disease-free survival (DFS) (*p* = 0.559). When comparing patients who met one or two Sedlis criteria, there were no statistically significant differences in OS between RT and no adjuvant treatment in either group. Since the selection of patients with cervical cancer eligible for surgery is becoming more accurate, adjuvant RT might not be necessary for patients with intermediate risk factors.

## 1. Introduction

Cervical cancer is the fourth most frequently diagnosed malignancy and the fourth leading cause of cancer-related death among women worldwide [1]. Despite recent advances in screening programmes and vaccines against the human papillomavirus, cervical cancer remains a global health concern. Treatment options for cervical cancer depend on the tumour stage at diagnosis, the existence of histopathological risk factors and the patient’s reproductive desires. In early-stage cervical cancer (stages IB1, IB2 and IIA1 according to the International Federation of Gynaecology and Obstetrics –FIGO 2018– classification), radical hysterectomy (RH) with pelvic lymph node assessment is the standard treatment. In these stages, radiotherapy (RT) has shown similar survival rates but is associated with more side effects, being therefore reserved for cases in which surgery is contraindicated [2,3]. The combination of radical surgery and RT is associated with significantly increased morbidity and should be avoided. Thus, the purpose of RH is to totally remove the tumour by resecting the uterus –including the cervix, parametrium, and the upper vaginal cuff– without requiring additional therapies. However, there are some circumstances in which the pathological post-operative outcome may make it necessary to associate an adjuvant treatment in order to reduce the possibility of recurrence. 

The Gynaecologic Oncology Group (GOG) [4] reported in 1990 that the presence of lymphovascular space invasion (LVSI), the depth of stromal invasion, and the size of the tumour are all independent prognostic factors for tumour recurrence after RH. Since then, patients presenting these pathological conditions are classified into an intermediate recurrence risk group. Based on these findings, in 1999 Sedlis et al. [4] published the results of a prospective randomised study (GOG-92 trial) comparing RT with no further treatment in patients presenting a combination of these intermediate risk factors (IRFs) after RH, reporting a 46% decrease in the recurrence rate for the RT group. These findings led numerous worldwide guidelines to suggest adjuvant RT as the preferred course of action for patients who met these intermediate risk criteria [3,5,6,7]. However, this study was performed more than two decades ago and its results may not be comparable with current clinical practice as the measurement of tumour size was carried out exclusively by clinical examination instead of imaging and surgical radicality was not described. Many more recent studies show that excellent local control can be achieved with surgery alone, although most of them are retrospective [8,9]. The need of RT in these intermediate-risk patients is, therefore, a controversial issue nowadays.

On the other hand, patients who present positive margins, microscopic involvement of the parametrium, or positive pelvic lymph nodes in the pathological analysis are thought to have a high risk of recurrence because it has been demonstrated that the presence of these conditions greatly lowers survival rates. This subgroup of patients benefits from concurrent administration of RT + chemotherapy (CT), as this approach presented greater overall survival (OS) and disease-free survival (DFS) rates in the prospective randomised trial conducted by Peters et al. [10] in 2000. Since then, patients with the aforementioned pathological findings have generally been advised to get this adjuvant treatment after RH [3].

The primary aim of this study was to determine whether adjuvant RT has an impact on survival after RH for early-stage cervical cancer in accordance with the Sedlis criteria. The secondary aim was to analyse if there is a significant difference in OS between patients presenting one or two IRFs depending on whether or not they received RT. We also evaluated whether there are other prognostic factors associated with tumour recurrence for these patients.

## 2. Materials and Methods

### 2.1. Study Design and Data Selection

We conducted a retrospective study including consecutive patients diagnosed with early-stage cervical cancer at the Gynaecological Oncology Unit of La Paz University Hospital between January 2005 and December 2022. All preoperative stages of the FIGO 2018 classification scheme (IA1 with LVSI, IA2, IB1, IB2 and IIA1) and all histologic subtypes were included in the study. Although RH is not currently recommended in IA stages, it was an accepted alternative in many guidelines when LVSI was present, so these patients were also included in our study. All patients underwent primary surgical treatment by RH with pelvic lymphadenectomy or pelvic sentinel lymph node biopsy (SLNB) and received follow-up care at the same centre. Exclusion criteria included patients undergoing fertility-sparing surgery or radical parametrectomy after simple hysterectomy, as well as patients with advanced stages or incomplete medical records. Querleu-Morrow’s classification was used to describe the radicality of the intervention [11]. All surgical procedures were performed by gynaecologic oncologists. External beam radiation (EBRT), CT and brachytherapy (BT) were selectively used postoperatively depending on the institutional tumour board’s decision and based on the presence of histologic risk factors and the patient’s characteristics. Irradiation of the pelvis was performed with a total dose of 45 to 50.4 Gy. When CT was administered, weekly cisplatin (40 mg/m^2^) was used as chemotherapeutic agent.

Tumour diameter ≥4 cm (in the final paraffin section), presence of LVSI and/or deep or middle third stromal invasion were defined as Sedlis criteria (intermediate risk) [4]. Peters criteria [10] (high risk) were defined as the presence of parametrial invasion, positive pelvic lymph nodes (micrometastases or macrometastases) and/or positive surgical margins. 

The data were obtained through a review of patients’ medical records after approval from the Ethics and Clinical Research Committee at the La Paz University Hospital (Reference PI-3668). Clinical, surgical and pathological data were collected from all eligible patients. Patients were classified into those with no adjuvant treatment and those who received adjuvant treatment. 

OS was defined as the time from the end of treatment to the date of death. DFS was defined as the time from the end of treatment to the diagnosis of recurrence (local or metastatic). 

### 2.2. Statistical Analysis 

A descriptive analysis of the quantitative variables was performed using the mean and standard deviation (SD) or the median and interquartile range (IQR) if the variables were not normally distributed; these variables were compared using Student’s *t*-test or the Mann–Whitney test in cases of non-normality. Qualitative variables were described using frequency distributions and percentages, and they were compared using the Pearson χ2 test or Fisher’s exact test. Kaplan-Meier survival analysis was used for the calculation of OS and DFS. Survival curves were compared using the log-rank test. Univariate and multivariate analyses of prognostic factors to predict survival outcomes were performed using Cox proportional-hazards regression models. Backward stepwise model selection was performed for variables with *p* ≤ 0.20 in univariate analysis, with *p* < 0.05 to be retained in the final model. Statistical significance was defined as a *p* value of less than 0.05. All statistical analyses were performed using STATA Statistics/Data Analysis (StataCorp LP, College Station, TX, USA).

## 3. Results

Of our initial cohort of 134 patients, a total of 121 (90.29%) eligible patients were included in the study. The median age in the cohort was 48.4 years (SD 11.5). Tumour types included squamous cell carcinoma in 74 patients (61.7%), adenocarcinoma in 18 patients (15%), adenosquamous carcinoma in 23 patients (19.2%) and other infrequent subtypes in 5 patients (4.2%). The most frequent FIGO stage was IB2 in 48 patients (40%), followed by IB1 in 32 patients (26.7%). Death data was missing for 6 patients. Of the remaining 115 patients, the median follow-up time was 70.2 months (IQR: 31.9 to 120.8). Recurrence rate was 24.79%. 73 patients (60.3%) underwent no adjuvant treatment and 48 (39.7%) received adjuvant RT (with or without CT). 56 patients (56.6%) presented one Sedlis criteria, of whom 46.4% received adjuvant RT, in contrast with patients who had two Sedlis criteria (26.3%), with 73% receiving RT. There were only 4 patients (4%) who met three Sedlis criteria and all of them received adjuvant RT. 

We formed two groups depending on whether patients received adjuvant treatment or not. The baseline, surgical and pathological characteristics of both groups are shown in Table 1. Patients who received RT treatment had more advanced tumour stages (*p* < 0.001), higher clinical (*p* < 0.001) and radiological (*p* < 0.0001) tumour size and lower conisation rates (*p* = 0.004). Moreover, the RT group had higher tumour grades (*p* < 0.001), increased LVSI (*p* < 0.001), a higher proportion of positive margins (*p* = 0.001) and higher parametrial (*p* < 0.001) and deep stromal (*p* < 0.0001) invasion rates. In terms of Sedlis criteria and Peters criteria, in the adjuvant treatment group there was a higher number of patients who met two Sedlis criteria (*p* = 0.002) or at least one Peters criteria (*p* < 0.001). 

After adjustment for statistically significant variables upon univariate analysis, radiotherapy was not found to be a statistically significant prognostic factor for OS (hazard ratio [HR] = 1.605, 95% confidence interval [CI]: 0.294–8.767, *p* = 0.584) or DFS (HR = 1.536, 95% CI: 0.364–6.482, *p* = 0.559). However, positive LVSI and rare tumour histologies (different from squamous cell carcinoma, adenocarcinoma and adenosquamous carcinoma) were correlated with OS (*p* = 0.019, *p* = 0.021 respectively), although not with DFS (*p* = 0.098, *p* = 0.654 respectively). Having prior conisation was a protective factor for recurrence (HR = 0.269, 95% CI: 0.080–0.898, *p* = 0.033). The number of positive pelvic lymph nodes, maximum tumour diameter, tumour grade, presence of positive margins, stromal invasion, parametrial invasion and tumour histology were not correlated with OS and DFS rates. The results of multivariate analysis are shown in Table 2. 

Finally, when comparing patients who met only one or two Sedlis criteria (and no Peters criteria), there were no statistically significant differences in OS between RT and no further treatment in either group (log-rank 0.112 and 0.8553, respectively) (Figure 1).

## 4. Discussion

Until recently, Sedlis criteria, based on a combination of tumour risk factors for recurrence, were used to determine whether adjuvant radiation following radical hysterectomy was appropriate for treating cervical cancer. However, more recent research questions the survival benefit of this adjuvant treatment, which makes this a topic of constant debate today. The European Society of Gynaecological Oncology (ESGO) feels that adjuvant RT should be taken into consideration in patients who meet a combination of IRFs, yet they recommend observation as an alternative option when an adequate type of RH is performed [3]. 

The results of our study showed that RT does not have an impact on survival in patients who meet one or two intermediate risk criteria yet do not have high risk criteria, and RT was not found to be a statistically significant prognostic factor for OS or DFS in multivariate analysis. 

As previously mentioned, the GOG-92 trial [4] was the one that determined the benefits of adjuvant RT in patients with IRFs after RH. In this study, patients that presented at least two IRFs were included. They reported a DFS rate at two years of 88% in the RT group and 79% in the no further treatment group. In the extended follow-up data published by Rotman et al. [12], the improvement in DFS rates was confirmed in the RT group. However, no significant differences in OS were found (HR = 0.70; 90%CI = 0.46–1.05; *p* = 0.074). Although this is a prospective randomised study, it was published more than 20 years ago and has many shortcomings [13]. 

First of all, tumour size was assessed by visual inspection rather than by imaging techniques as recommended by current quality indicators for the surgical treatment of cervical cancer [14]. Secondly, the parametrial resection type was not reported, making it impossible to determine the extent of radicality–and this is essential data to assess if optimal surgical management was achieved. Moreover, 26.7% of the patients in the Sedlis study had tumours >4 cm that would probably not be candidates for surgery today. Furthermore, SLNB ultrastaging performed nowadays can exclude high-risk patients that may not have been detected in the GOG-92 trial, which would result in significant bias, since they would have been considered to present an intermediate risk of recurrence rather than a high risk. All these weaknesses could mean that the more precise preoperative staging performed today could identify extrauterine disease with more accuracy than in the population of the Sedlis trial; thus, the results of said trial cannot be extrapolated to today’s clinical practice [13]. 

Studies conducted after the GOG-92 trial are mostly retrospective and the results are contradictory. Schorge et al. [15] reported that patients with a presence of LVSI who received adjuvant RT developed less recurrences that those who received surgery alone (*p* = 0.04). A Cochrane meta-analysis published in 2012 concluded that adjuvant RT decreases DFS without improving OS [16]. However, this meta-analysis was based on only two trials (the GOG-92 trial and another one published in 1982), limiting the validity of the results. 

In contrast to the abovementioned studies, much more recent ones show that adequate surgical management may be sufficient for intermediate-risk patients after RH. In the study carried out by Haesen et al. [17] that included 182 patients with a FIGO stage of IB1 and a combination of IRFs, the recurrence rate was only 10% despite the fact that most of the patients did not receive adjuvant RT. In the large retrospective study of 765 IRF patients performed by Nasioudis et al. [18], no benefit in OS (*p* = 0.44) was found from adjuvant RT (±CT), even after controlling for age, histology and surgical approach. Other retrospective studies with smaller cohorts of patients found similar results, with no survival benefits from adjuvant RT or concurrent chemoradiotherapy when compared to no further treatment [19,20]. 

Different statistical methods were employed in various retrospective studies to homogenise the groups and increase the reliability of the findings, which led to similar conclusions: in the multicentric study performed by Ye et al. [8] that included patients with only one IRF and compared adjuvant RT, adjuvant CT and no adjuvant treatment, no significant differences were found in the 5-year OS (*p* = 0.486) and DFS (*p* = 0.874) rates between the RT group and the no further treatment group after propensity score matching; similar results were recently found by Tuscharoenporn et al. [21] in their propensity score-adjusted analysis of 219 patients with stages IB-IIA and IRF according to the criteria of the Sedlis trial. There was no significant association between adjuvant treatment and OS (*p* = 0.36) or DFS (*p* = 0.42), although a lower pelvic relapse rate was observed in the RT group (*p* = 0.02). 

In contrast to these results, in the meta-analysis performed by Sagi-Dain et al. [22], adjuvant RT showed a reduced recurrence risk in terms of OS for patients with two IRFs (OR 1.86, *p* = 0.04), but these benefits were not found after adding patients with only one risk factor. 

Since all these studies differ from each other in the number of Sedlis criteria included, and that many of them indicate adjuvant RT when, at least, two criteria are met, we investigated whether there was a benefit in terms of OS in our cohort in patients who received RT depending on whether they presented only one or two IRFs (and no high risk factors). No statistically significant differences were found in either of the two groups. According to our results, in the study of 134 patients with one or more IRFs [23] the number of positive IRFs did not affect DFS (5-year DFS of 84.7% for one IRF and 5-year DFS of 85.6% for two or three IRFs; *p* = 0.994). The effect on OS could not be calculated because only 5 patients died during the follow-up period. 

In the most recent meta-analysis performed by Gómez-Hidalgo et al. [24] that included eight studies (the Sedlis trial being among them), similar outcomes in terms of recurrence and mortality between adjuvant treatment and observation were reported. 

Following the results obtained in two retrospective studies conducted by Cibula et al. [25,26] in which radical surgery alone achieved similar OS and DFS rates compared to those achieved by adjuvant treatment, a prospective randomised trial has been developed (the CERVANTES trial) which is currently ongoing and is expected to provide definite insights into the role of radiotherapy in patients with intermediate-risk cervical cancer [27]. 

Our study had several limitations, mainly due to its retrospective nature and the limited number of patients. Since most of the patients who received adjuvant treatment had poor prognostic factors, both treatment groups were not completely balanced. In addition, our study included all histological subtypes, compared to most publications which exclude the infrequent ones–and this factor could have generated bias. Likewise, the imaging technique used for diagnosis prior to 2009 at our centre was ultrasound instead of magnetic resonance imaging. This may have increased the number of patients who were not actually candidates for RH. Despite all this, RT has not been shown to be a statistically significant prognostic factor for recurrence or death in the multivariate logistic regression model and OS was not improved in patients who had only one or two IRFs and received RT when compared with the no further treatment group.

## 5. Conclusions

No benefits in terms of survival were found in patients with two Sedlis criteria who received RT, which is in line with the most recent results in the literature. It is possible that the improvement of diagnostic and surgical techniques in recent years may allow for more accurate selection of patients eligible for surgery, permitting optimal control of the disease without the need for adjuvant treatments. The results of prospective randomised trials are needed to confirm these findings.

## Figures and Tables

**Figure 1 jpm-13-01486-f001:**
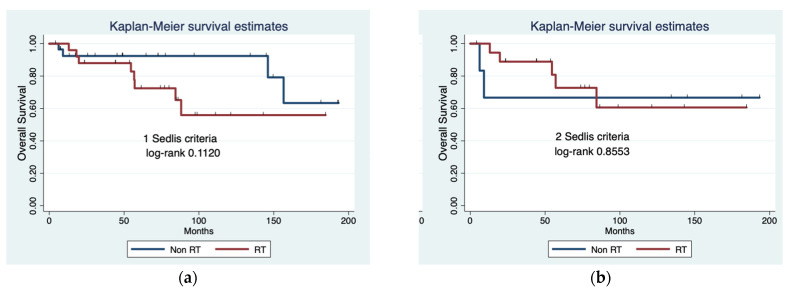
Overall survival (OS) comparing radiotherapy (RT) versus no adjuvant treatment in patients who met one (**a**) or two (**b**) Sedlis criteria.

**Table 1 jpm-13-01486-t001:** Baseline, surgical and pathological characteristics from patients (n = 121) from different treatment groups. Data are given in median [interquartile range] or frequencies (relative percentages). BMI: body mass index. SCC: squamous cell carcinoma. AC: adenocarcinoma. ADSC: adenosquamous carcinoma. SLNB: selective lymph node biopsy.

	Non RT Group n = 73	RT Group n = 48	*p* Value
Age (years)	47.9 (11.3)	49.1 (11.8)	0.5864
BMI (kg/m^2^)	27.0 (6.2)	25.9 (5.0)	0.3142
FIGO stage (2018) IA1 IA2 IB1 IB2 IB3 IIA1 IIA2			<0.001
	2 (2.7%)	0	
	6 (8.2%)	0	
	30 (41.1%)	5 (10.4%)	
	34 (46.6%)	32 (66.7%)	
	1 (0.4%)	5 (10.4%)	
	0	5 (10.4%)	
	0	1 (2.1%)	
Clinical tumor size (measured by eye) Microscopic ≤2 cm >2 cm			<0.001
	33 (48.5%)	8 (19.1%)	
	12 (17.7%)	2 (4.8%)	
	23 (33.8%)	32 (76.2%)	
Maximum size per image (mm)	12.5 (13.0)	27.9 (16.6)	<0.0001
Conization performed	41 (56.2%)	14 (29.2%)	0.004
Tumoral size in cone (mm)	15.5 (12.1)	21.5 (9.3)	0.1397
Type of parametrial resection A B C1 Missing data			0.014
	2 (2.7%)	0	
	29 (39.7%)	9 (18.8%)	
	41 (56.2%)	38 (81.3%)	
	1 (1.4%)	0	
Surgical approach Open Laparoscopic Robotic			0.871
	13 (18.8%)	8 (16.7%)	
	59 (80.8%)	40 (83.3%)	
	1 (1.4%)	0	
SLNB	50 (68.5%)	32 (66.7%)	0.833
Bilateral pelvic lymphadenectomy	40 (54.8%)	39 (81.2%)	0.003
Number of positive lymph nodes	0 (0)	0 [IQ 25–75: 0–1] 0.5 (1.0)	<0.0001
Postsurgical FIGO stage IA1 IA2 IB1 IB2 IB3 IIA2 IIB IIIC1			<0.001
	4 (5.6%)	0	
	4 (5.6%)	0	
	29 (40.3%)	3 (6.3%)	
	29 (40.3%)	19 (39.6%)	
	6 (8.3%)	7 (14.6%)	
	0	1 (2.1%)	
	0	5 (10.4%)	
	0	13 (27.1%)	
Maximum tumor diameter (mm)	21.2 (12.9)	32.4 (14.5)	<0.0001
Histology SCC AC ADSC Other			0.214
	40 (55.6%)	34 (70.8%)	
	11 (15.3%)	7 (14.6%)	
	18 (25.0%)	5 (10.4%)	
	3 (4.2%)	2 (4.2%)	
Tumor grade 1 2 3			<0.001
	26 (37.1%)	3 (6.3%)	
	34 (48.6%)	24 (50.0%)	
	10 (14.3%)	21 (43.8%)	
Positive LVSI	6 (8.5%)	31 (66.0%)	<0.001
Positive margins	0 (0%)	7 (14.6%)	0.001
Depth of stromal invasion (mm)	7.3 (4.4)	14.1 (6.9)	<0.0001
Stromal invasion in thirds Inner 1/3 Middle 1/3 Deep 1/3			<0.001
	25 (45.5%)	1 (2.2%)	
	17 (30.9%)	9 (19.6%)	
	13 (23.6%)	36 (78.3%)	
Parametrial invasion	0 (0%)	8 (16.7%)	<0.001
Vaginal invasion	0 (0%)	3 (6.3%)	0.060
1 Sedlis criteria	30 (57.7%)	26 (55.3%)	0.812
2 Sedlis criteria	7 (13.5%)	19 (40.4%)	0.002
3 Sedlis criteria	4 (4%)	1 (1.9%)	0.343
At least 1 Peter criteria	0 (0%)	21 (43.8%)	<0.001

**Table 2 jpm-13-01486-t002:** Cox multivariate survival analysis. OS: overall survival. DSF: disease-free survival. HR: hazard radio. CI: confidence interval. LVSI: lymphovascular space invasion.

Prognostic Factors	OS			DFS		
	HR	*p*	95% CI	HR	*p*	95% CI
Tumor diameter	1.036	0.271	0.972–1.104	1.017	0.433	0.973–1.063
Previous conization	-	-	-	0.269	0.033	0.080–0.898
Grade	1.985	0.371	0.442–8.921	1.830	0.306	0.574–5.826
Positive LVSI	6.024	0.019	1.339–27.091	2.658	0.098	0.834–8.473
Positive Margins	1.266	0.929	0.007–222.530	1.839	0.710	0.073–45.921
Stromal invasion	1.079	0.100	0.985–1.181	0.997	0.948	0.912–1.089
Parametrial invasion	11.074	0.372	0.056–2169.149	1.558	0.779	0.070–34.551
Number of pelvic lymph nodes	2.539	0.082	0.887–7.262	1.796	0.154	0.802–4.022
Histology Adenocarcinoma Adenosquamous Other						
	4.155	0.064	0.919–18.783	1.153	0.841	0.285–4.663
	-	.	.	1.456	0.566	0.403–5.256
	9.129	0.021	1.387–60.076	1.671	0.654	0.177–15.778
Radiotherapy	1.605	0.584	0.294–8.767	1.536	0.559	0.364–6.482

. The incidence of adenosquamous caricnoma was so low, that the OS couldn’t be calculated.

## Data Availability

The data presented in this study are available on request from the corresponding authors. The availability of the data is restricted to investigators based in academic institutions.

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
