# Peer review of "Role of Adjuvant Radiotherapy in Patients with Cervical Cancer Undergoing Radical Hysterectomy"

_jpm, 2023, doi:10.3390/jpm13101486_

Round 1

Reviewer 1 Report

I have gone through the manuscript. Topic is interesting but authors have missed many aspects to realistically analyze about the efficacy of radiotherapy. 

Authors did not discuss about the side-effects. Analysis of side-effects should be discussed for a better overall analysis. 

Did the authors also treat metastasis patients? Did the authors find regression of the secondary tumor Growth as well?

The authors should discuss the effects of radiotherapy on Primary tumors, Lymph node metastasis and secondary tumors

Moderate editing of English language required

Author Response

Dear reviewer, please find next all the answers to your comments.

As suggested, we have sent our manuscript to an academic editing platform for professional English improvement

In addition, we have expanded the length of the article.

Reviewer question 1: Authors did not discuss about the side-effects. Analysis of side-effects should be discussed for a better overall analysis.

Answer: Dear reviewer. Unfortunately, as this study is focused on patients with radical hysterectomy, the side effects collected are those secondary to the surgical intervention, and we do not have those associated with radiotherapy treatment. We greatly appreciate this comment and we will take it into account for future studies.

Reviewer question 2: Did the authors also treat metastasis patients? Did the authors find regression of the secondary tumor Growth as well?

Answer: Dear reviewer, patients with metastases were excluded from our study since only those patients with early-stage cervical cancer were included.

In the follow-up of these patients, there were patients with recurrences who received surgery, radiotherapy, chemotherapy or a combination of them, depending on the type of recurrence or metastatic pattern. However, the objective of this study was to analyze the value of radiotherapy after radical hysterectomy, so the treatment of metastases was not studied.

Reviewer question 3: The authors should discuss the effects of radiotherapy on Primary tumors, Lymph node metastasis and secondary tumors.

Answer: Dear reviewer, thank you for your comment. The objective of this study was to evaluate the role of radiotherapy exclusively after radical hysterectomy, in patients with early-stage cervical cancer, since its value is controversial today. The role of radiotherapy for advanced stages is widely demonstrated, so patients with metastases or recurrences were not included in this study.

Reviewer 2 Report

In the current study, the authors have investigated the consequences of adjuvant radiotherapy on the clinical outcome of cervical cancer cases that performed radical hysterectomy. The study is well-designed and I have some comments for the authors:

1- In the introduction section, authors should add a paragraph mentioning the concept of adjuvant and neoadjuvant therapy that showed successful results in different human cancers.

2- The first paragraph of the discussion section should start with an introductory sentence instead of starting directly with (The results of this study showed .....).

3- It would be interesting if the authors discussed other reports that analyzed adjuvant/neoadjuvant therapeutic approaches (chemotherapy or immunotherapy) in patients with radical hysterectomy. If such studies are not reported in the literature, suggesting such approaches as future steps may improve the current study discussion section. 

Minor editing of English language required

Author Response

Dear reviewer, please find next all the answers to your comments.

As suggested, we have sent our manuscript to an academic editing platform for professional English improvement

In addition, we have expanded the length of the article.

Reviewer comment 1: In the introduction section, authors should add a paragraph mentioning the concept of adjuvant and neoadjuvant therapy that showed successful results in different human cancers.

Answer: Dear reviewer, thank you for your comment. In order to keep the focus of the article on cervical cancer and radiotherapy, we do not feel that a so unspecific paragraph will add value to the paper.

Reviewer comment 2: The first paragraph of the discussion section should start with an introductory sentence instead of starting directly with (The results of this study showed…)

Answer: Dear reviewer, thank you for this comment. We have modified the start of the discussion section and we have included an introductory paragraph.

Reviewer comment 3: It would be interesting if the authors discussed other reports that analyzed adjuvant/neoadjuvant therapeutic approaches (chemotherapy or immunotherapy) in patients with radical hysterectomy. If such studies are not reported in the literature, suggesting such approaches as future steps may improve the current study discussion section.

Answer: Dear reviewer, thank you for your comment. The standard treatment for cervical cancer in early stages is radical hysterectomy, although primary radiotherapy can be considered since it has similar survival rates. However, treatment with chemotherapy/immunotherapy has no value in the initial stages of the disease and is only used in patients with locally advanced or metastatic stages, so we consider that it is not interesting to mention it in this article since in our study we only included patients with early-stage cervical cancer who underwent surgical treatment.

Round 2

Reviewer 1 Report

Looks in better form now. 

 Moderate editing of English language required

Reviewer 2 Report

The manuscript can be accepted in the present form

Minor editing of English language required